# Is the Use of Segways or E-Scooters in Urban Areas a Potential Health Risk? A Comparison of Trauma Consequences

**DOI:** 10.3390/medicina58081033

**Published:** 2022-08-02

**Authors:** Kai Hoffeld, Olivia Mair, Markus Wurm, Philipp Zehnder, Dominik Pförringer, Peter Biberthaler, Chlodwig Kirchhoff, Michael Zyskowski

**Affiliations:** 1Department for Trauma Surgery, Klinikum Rechts der Isar, Technichal University Munich, 81675 Munich, Germany; olivia.mair@mri.tum.de (O.M.); markus.wurm@mri.tum.de (M.W.); philipp.zehnder@mri.tum.de (P.Z.); dominik.pfoerringer@mri.tum.de (D.P.); peter.biberthaler@mri.tum.de (P.B.); chlodwig.kirchhoff@mri.tum.de (C.K.); 2Department for Orthopedics and Trauma Surgery, University Clinic Cologne, 50937 Cologne, Germany

**Keywords:** e-scooter, Segway, electromobility, injury patterns, traffic incidents, traffic safety

## Abstract

*Background and objectives:* Electromobility has become increasingly popular. In 2001, Segway personal transporters (Segway) were established for tourists, and e-scooters have been in use since their approval in 2019. The aim of this study was to analyze and compare the types of injuries directly related to the use of Segways and e-scooters, respectively, in a German city and to phrase potential safety advice. *Materials and Methods:* All patients presenting to our emergency department after Segway incidents were retrospectively analyzed and compared with the prospectively collected cohort of patients following e-scooter incidents. Presented injuries were analyzed by body region and injury severity score (ISS). Epidemiological data were collected. *Results:* Overall, 171 patients were enrolled. The Segway group included 56 patients (mean age 48 years), and the e-scooter group consisted of 115 patients (mean age 33.9 years). Head injuries (HI) occurred in 34% in the Segway group compared to 52% in the e-scooter group. The ISS was approximately equal for both groups (mean ISS Segway group: 6.9/e-scooter group: 5.6). *Conclusions:* Since the e-scooter group presented a high number of HI along with a higher likelihood and greater severity of HI, mandatory use of helmets is suggested.

## 1. Introduction

Electromobility changed the nature of private transport in the last years and has become increasingly important in urban inner cities around the world. Since the “Electric Mini Vehicles Ordinance” came into force on 15 June 2019, electrically powered pedal scooters (“e-scooters”) have been permitted on German inner city roads [1]. In Munich, several thousand e-scooters are offered by various providers for use in individual transport [2]. Segway personal transporters (Segway) were developed to improve individual transport with the intention of changing urban mobility according to the inventors’ ideas [3]. This aspiration was never satisfied; in the end, the Segway has primarily become a tourist attraction. Most major cities offer guided tours for comfortable tourist touring [4].

The vehicle is operated by the “lean steer technology”, enabling the driver to stand on the platform in between two wheels and to choose directions by leaning toward it. The Segway is equipped with sensors and accelerometers to balance the vehicle by providing opposing forces to counteract gravity. This e-vehicle can reach 20 km/h and is able to travel 38 km by charging it once [5]. In comparison, the e-scooter also holds two wheels and a platform on which the driver stands during the ride. Steering maneuvers are carried out through a handlebar as well. The speed control lever is located on the handlebar. Depending on the model, the e-scooter can travel at speeds of up to 45 km/h, although a maximum speed of 20 km/h is allowed in German road traffic. Depending on the battery, the vehicle has a range of up to 130 km, but on average, about 40–60 km are reached with a single load-up [6].

A review of the literature revealed that Segway incidents result in serious injuries and even demanding operative treatment, which is associated with increased morbidity and high medical costs [7,8,9]. Recent international studies showed that the use of e-scooters in urban areas presents a safety risk and is associated with severe injuries such as traumatic brain injuries (TBI) and fractures of the skull, face, and limbs [10,11,12,13,14,15]. At the same time, these studies report a lack of helmet use in e-scooter riders, suggesting an association with a higher risk and severity of head injuries [10,11,12,13,14,15].

Due to the fact that both vehicles are almost only available for rental and there is no need for physical effort due to the electric character of the vehicles, we see a strong similarity between Segways and e-scooters.

The aim of this study was to analyze and compare the types of injuries directly related to the use of Segways and e-scooters in a major German city with over a million inhabitants. In addition, the need for medical care and the associated circumstances such as driving under the influence of alcohol, time of day, and weekday were investigated. 

## 2. Materials and Methods

This is a retrospective case series with a level IV level of evidence. Institutional Review Board approval was obtained prior to this study (IRB approval no: 485/21 S-NP, Ethical Committee of Technical University Munich). All patients’ data from the emergency department of our Level I University Hospital related to either the use of a Segway or an e-scooter were retrospectively analyzed. All person-related data were anonymized. For this purpose, the hospital software (SAP) was searched for the keywords “Segway”, “electric scooter”, and “e-scooter”. Patients were categorized into two groups according to the vehicle used: the Segway cohort and the e-scooter cohort. Data for the Segway group were collected during the period between 1 January 2010 and 31 March 2021. The data for the e-scooter group were collected from 1 July 2019 to 31 March 2021. The Segway group and the e-scooter group were analyzed for injury patterns and severity using the injury severity score, ISS. Furthermore, an evaluation of age, sex, alcohol influence, use of protective gear, and time during the day and weekday of the incident was performed. In addition, the kind of treatment and the need for inpatient admission were investigated. The relevant information was obtained from the electronic medical records. If the documentation was not clear, missing information was scored as “unknown.” When detailed information for the variables “alcoholized patient” and “use of protective gear” was not given in the available documents, the variable “unknown” was set. In these cases, “unknown” was scored as non-alcoholized patient and no use of protective gear, respectively. Information on whether a patient was under the influence of alcohol was obtained by patients reporting whether and how much alcohol they had drunk, in addition to obvious alcoholic behavior. Further tests for blood alcohol were not performed. The use of other drugs was not observed nor reported.

A *p*-value of 0.05 was set as statistically significant. Accordingly, confidence intervals were calculated at 95%. Continuous variables have been summarized by the mean and standard deviation (SD). To check for normal distribution, the Kolmogorov–Smirnov test was used. For non-normally distributed samples, the Wilcoxon–Mann–Whitney test was further used. Only the age in the Segway group was normally distributed. All other subsets were non-normally distributed. Categorical variables were compared using the chi-square test. When data series were correlated, the Pearson correlation coefficient and the corresponding significance level of the Pearson correlation were calculated. The statistical calculations have been performed using the IBM SPSS Statistics for Windows, Version 26.0. Armonk, NY: IBM Corp.

## 3. Results

### 3.1. Epidemiology

In the evaluation time period between 01/01/2010 and 31/03/2021, a total of 171 patients were retrospectively included and evaluated in this study. In the Segway group (I), 56 (34 men and 23 women) injured patients were enrolled compared to 115 (66 men and 49 women) patients in the e-scooter group (II). Every e-scooter driver stated that he or she was riding a rented e-scooter. Only one Segway driver stated that he was riding his own Segway. There was no statistically significant difference in the sex distribution (*p* = 0.679). The mean age in the Segway group was 48 years (SD ± 14.1 years), which was significantly higher (*p* ≤ 0.001) than the mean age of 33.9 years (SD ± 13.4 years) in the e-scooter group. Regarding the condition of driving under the influence of alcohol, not a single patient in the Segway group was driving drunk on a Segway. In comparison, 33% of the e-scooter group patients had an incident under the influence of alcohol. (*p* ≤ 0.001). No statistically significant difference was found for the weekday of the incident. In 43% of the cases in the Segway group, the patients’ presentation occurred during the week (Monday–Thursday) and 57% on weekends (Friday–Sunday). A similar distribution was observed in the e-scooter group, with 42% of the incidents occurring during the week and 58% on weekends (*p* = 0.889). When considering the most likely time of an incident, differences were found in both groups. In the Segway group, incidents most likely occurred during the daytime between 3:00 pm and 9:00 pm (48%, *p* = 0.008), whereas patients of the e-scooter group most likely (43%) had an incident at night between 9:00 p.m. and 07:00 a.m. (*p* ≤ 0.001). 

A helmet was worn by at least 48% of the injured patients in the Segway group but only by 1.7% of the e-scooter riders (*p* ≤ 0.001). Incidents among tourists were significantly higher in the Segway group, 54%, than in the e-scooter group (33%, *p* = 0.01). Commuting incidents did not occur at all in the Segway group, while 5% of incidents in the e-scooter group occurred while commuting to or from work (*p* = 0.082). For details, see Table 1.

### 3.2. Injury Pattern

Significant differences in injury patterns were particularly evident in injuries to the head. Head injuries occurred in 34% of the Segway group and in 52% of the e-scooter group (*p* = 0.025). Similar data were found regarding fractures of the skull. The e-scooter group showed 21% (*n* = 24) of the patients suffering from fractures of the skull presenting a significantly higher rate than in the Segway group with 4% (*p* ≤ 0.001). Fractures of the facial skull occurred in 4% of the Segway group and in 21% of e-scooter-patients (*p* ≤ 0.001). Soft tissue injuries of the head were significantly higher in the e-scooter group. We observed 11% soft tissue injuries in the Segway group and 29% in the e-scooter group (*p* = 0.009). Tooth fractures did not occur in any Segway, but in 8% of e-scooter riders (*p* = 0.031). Nevertheless, no significant difference was found between the two analyzed groups regarding the incidence of traumatic brain injuries (Segway group: 18%, e-scooter group: 31%, *p* = 0.063). Intracerebral bleeding was found in 5% of the Segway group and in 4% of the e-scooter group (*p* = 0.76). 

In general, both groups showed high fracture rates. Our data revealed that 52% of the patients in the Segway group and 44% of the patients in the e-scooter group suffered from a fracture (*p* = 0.36). Injury patterns of the upper and lower extremities and trunk were similar in both groups. Injuries of the upper extremity occurred more often (Segway group: 50%, e-scooter group: 38%, *p* = 0.145) compared to lower extremity injuries (Segway group: 29%, e-scooter group: 34%, *p* = 0.483). The most common upper extremity fractures were distal radius fractures in the Segway group (7%, *n* = 4) and radial head fractures in the e-scooter group (9%, *n* = 10). The most common lower extremity fracture in both groups were tibial plateau fractures (Segway group: 7%, e-scooter group: 3%, *p* = 0.287). Bruises or fractures of the ribs were the most common injury to the trunk. Minor injuries (i.e., contusions) were observed in 34% (*n* = 19) in the Segway group and in 38% (*n* = 44) in the e-scooter group. Abrasions and lacerations occurred at significantly higher rates in the e-scooter group with 48% (*n* = 55) of the cases, while these injuries were present in the Segway group in 27% (*n* = 15) of the cases (*p* = 0.009). In 3% of the cases, only a low count of joint dislocations was observed in both groups (*p* = 0.975). Overall, injury severity was approximately equal in both groups, with a mean ISS of 6.9 (range: 1–34) in the Segway group and 5.6 (range: 1–50) in the e-scooter group (*p* = 0.211). A positive correlation of ISS with age resulted for both groups (Pearson-r = 0.23). The correlation of ISS with the age of exclusively non-alcoholized patients across both groups presented an even clearer positive relationship with a Pearson-r of 0.36 (*p* ≤ 0.001) (see Table 2 and Table 3 and Figure 1).

### 3.3. Diagnostics

For diagnostic purposes, conventional X-ray diagnostics were performed in 71% of the Segway group and in 63% of the e-scooter group (*p* = 0.255). Regarding further computed tomography (CT) diagnosis of the extremities, a significant difference was recognized between the two groups. In 34% of the Segway group cases and in only 10% of the cases in the e-scooter group, a further CT scan was indicated (*p* ≤ 0.001). A CT scan of the head and cervical spine was obtained in 27% of the presented cases in the Segway group and in 36% of the e-scooter group (*p* = 0.246). Further computed tomography of the facial skull had shown a significant difference and was performed almost eight times more often in e-scooter riders (Segway group: 4%, e-scooter group: 30%, *p* ≤ 0.001). See Table 4.

### 3.4. Therapy

During primary care in the emergency room, significantly more wound care had to be performed in the e-scooter group (30%) than in the Segway group accounting for only 10% (*p* = 0.005). A splint or cast was applied in 30% of injured patients in the Segway group and in 29% of the e-scooter group (*p* = 0.823). Indication for surgical treatment accounted for 35% in the Segway group and for 25% in the e-scooter group (*p* = 0.154). See Table 4.

### 3.5. Discharge Modality

Regarding inpatient admission, immediately after presentation in the emergency department or in the further course of treatment after a performed surgical procedure, a significantly higher number of patients of the Segway group had to be hospitalized compared to e-scooter drivers (Segway group: 41%, e-scooter group: 23%, *p* = 0.017). See Table 4.

### 3.6. Subgroup Analysis

In a subgroup analysis, “intoxicated e-scooter riders” (subgroup I) were compared to “not-intoxicated e-scooter riders” (subgroup II), revealing significant differences in the severity of the injuries and inpatient admission rate. In detail, e-scooter riders in subgroup I showed higher ISS scores (7.9 vs. 4.5, *p* = 0.004) as well as a significantly higher rates of head injuries (subgroup I: 94%, *n* = 34 vs. subgroup II: 34%, *n* = 26; *p* ≤ 0.001). All observed intracranial hemorrhages (*n* = 5) occurred only in subgroup I. Furthermore, patients under the influence of alcohol were more likely to sustain a fracture (subgroup I: 58%, *n* = 21 vs. subgroup II: 42%, *n* = 32; *p* = 0.095) as well as wounds (subgroup I: 69%, *n* = 25 vs. subgroup II: 42%; *n* = 32; *p* ≤ 0.001). These results were accompanied by more frequent inpatient admissions in subgroup I (39%, *n* = 14. vs. subgroup II: 18%, *n* = 14; *p* = 0.018). In addition, subgroup I patients were at higher risk for surgical treatment (33%, *n* = 12; subgroup II: 18%, *n* = 14; *p* = 0.075). Injuries to the upper (subgroup I: 25%, *n* = 9 vs. subgroup II: 45%, *n* = 35; *p* ≤ 0.001) and lower extremities (subgroup I: 17%, *n* = 6 vs. subgroup II: 43%, *n* = 33 *p* ≤ 0.001) were less frequently represented in the group of drunk e-scooter riders compared to the non-alcoholized ones.

Diagnostic procedures in the two analyzed subgroups differed significantly. In subgroup I, significantly fewer X-rays were performed (subgroup I: 27%, *n* = 10 vs. subgroup II: 82%, *n* = 63; *p* ≤ 0.001). Regarding CT diagnostics, a reversed distribution occurred. In subgroup I, significantly more CT-scans of the head and cervical spine were performed compared to subgroup II (subgroup I: 67%, *n* = 24 vs. subgroup II: 22%, *n* = 17; *p* ≤ 0.001). In addition, significantly more often a CT scan of the facial skull was needed (subgroup I: 64%, *n* = 23 vs. subgroup II: 14%, *n* = 11; *p* ≤ 0.001). Another difference between both subgroups was the sex distribution. In subgroup I, 81% of the patients were male compared to only 47% of the subgroup II patients (*p* ≤ 0.001). See Table 3.

## 4. Discussion

The introduction of electromobility for private transport, especially in big cities, changed its nature in the last years and became increasingly important in urban inner cities around the world. In this context, the presented study compares for the first time the injury patterns occurring in e-scooter riders to another completely electrical-powered means of transportation, the Segway. In addition, the presented data represent the currently largest study population of e-scooter incidents in Germany. The comparison is obvious, as both vehicles are mostly used on a rental basis. Because of the more sporadic use of both vehicles, a training effect seems to be irrelevant. The Segway drivers were of greater age compared to e-scooter drivers (mean age Segway group 48.04 years vs. e-scooter group 33.9 years; *p* ≤ 0.001), which was simultaneously associated with a slightly higher mean injury severity score (ISS 6.9 in the Segway group 6.9 vs. 5.6 in the e-scooter group). When correlating these two variables, there was a positive correlation between age and ISS (Pearson-r = 0.23). When excluding “alcoholized patients” in the correlation of age and ISS, the result is an even clearer positive relationship between those two factors (Pearson-r 0.36). Furthermore, Segway drivers were significantly more often hospitalized than e-scooter drivers (Segway group 41% vs. e-scooter group 23%; *p* = 0.018). This fact may be related to the higher average age of the Segway group. As demonstrated, the higher age was accompanied by more severe injuries (see Figure 1) with an increased indication for direct inpatient admission. Because of commonly used antithrombotic medication, neurological supervision was often necessary for elderly patients suffering from TBI in a stationary setting [16]. In general, body control and bone quality decrease with advancing age [17,18]. It is known that age-associated changes in strength and balance, as well as lower bone quality, result in increased susceptibility to fractures and thus could contribute to increased injury severity in the Segway group [17,18,19,20]. Especially considering the more complex operation of the Segway by weight shifting compared to the intuitive operation of the e-scooter, body control seems to play an essential role in the cause of injury in the Segway group patients. Looking at the injury patterns in detail, several facts are noteworthy. Especially striking is the large difference in relevant head injuries since our data showed only 34% (*n* = 19) of Segway riders suffering from head injuries compared to 52% (*n* = 60) in the e-scooter group. This significant (*p* = 0.024) higher number of head injuries in the e-scooter group is comparable to results from recent studies in the current literature investigating e-scooter incidents. Mair et al. found head injuries in 52% of their cases, while Kleinertz et al. observed 54% and Heuer et al. 34% head injuries in e-scooter riders [10,11,14]. Shichman observed 17.4% of head injuries, which was lower than in the other mentioned studies but still the most common “non-orthopedic” injuries in that study [15]. In our study, only a moderate number of head injuries were detected in the Segway group, as described in similar studies investigating Segway injuries [7,8]. The association between head injuries and wearing a safety helmet is obvious. In this context, our findings underline several studies in the current literature describing that wearing a helmet is a crucial safety tool in 2-wheeled vehicle incidents [21,22,23]. In our study, 48% of the Segway group patients wore a helmet. Among patients in the e-scooter group, in just two cases (1.7%) was helmet use documented. This rarely observed use of safety helmets while riding an e-scooter is congruent with other studies on e-scooter injuries [10,11,13,14,15]. As already discussed, a lack of helmet use correlates with head injuries of all severity grades; this thus results in a recommendation for wearing a safety helmet for e-scooter riding. Regarding the presented results also, the influence of alcohol on the incidence and severity of head injuries, respectively, should be stressed. Our subgroup analysis of alcoholized (subgroup I) versus non-alcoholized e-scooter riders (subgroup II) showed a rate of 94% head injury in subgroup I and only 34% in subgroup II. This is supported by the study of Kleinertz et al., in which 92% of alcohol-intoxicated e-scooter riders sustained a head or facial injury. Similar results were described by Mair et al., who performed a subgroup analysis on e-scooter injuries during the Octoberfest in Munich and found that alcohol intoxication puts the drivers at high risk for severe injury with an average ISS of 6.7 (range 1–24) [10]. The results of our study are consistent with the findings of the mentioned studies. For example, all cases of intracranial hemorrhages observed in the e-scooter group were found in the alcoholized subgroup.

Having a closer look at limb injuries, a striking finding is that Segway patients sustained upper extremity injuries much more often than patients following e-scooter injuries (*p* = 0.145). The upper extremity was injured in 50% of the Segway group, but only in 38% of the e-scooter group, but also only in 25% of the subgroup I patients (*p* ≤ 0.001). Interestingly non-alcohol-intoxicated (subgroup II) e-scooter drivers presented with nearly the same number of upper extremity injuries as described for the Segway group patients following Segway injuries. The significant difference regarding injuries of the upper extremity between sober Segway riders and intoxicated e-scooter users is most likely due to the fact that alcohol leads to a reduced ability of anticipation resulting in a reduction in the protective reflexes. In our opinion, this is the main reason for the observed significant difference between upper extremity and head injuries in the two study groups. In general, it needs to be stated that alcohol had an overall negative impact on patient outcomes. For example, drunk patients presented with a higher average injury severity score (7.9 vs. 4.5). They furthermore required more often wound care (69% vs. 42%) and also suffered from a higher incidence of fractures (58% vs. 42%). In addition, surgical treatment was needed more often (33% vs. 18%), and they had to be hospitalized more often (39% vs. 18%). This is also coherent with other studies. For instance, Shichman registered increased injury rates during the late-night hours on weekends, which was attributed to alcohol involvement [15].

In Germany, the legislative authority prescribes a maximum speed limit of 20 km/h for the use of mini electric vehicles. A minimum age of 14 years is required for the use. There is no legislative obligation to wear a helmet. Regarding alcohol intoxication, the same alcohol per mile limit applies to car drivers, i.e., <0.5 per mile [24]. By definition, these regulations apply to the use of Segways and e-scooters. Segways are mainly offered in tourist operations for guided tours or individual rentals. Here, helmet use and alcohol abstinence are usually required. In contrast, e-scooters can be rented via a smartphone app. A corresponding control authority checking for alcohol abstinence or helmet use is missing. Consequently, helmets are very uncommonly worn by e-scooter users. We consider this the main reason for the observed differences in head injuries and severity of the head injuries between both analyzed groups. We hypothesize that e-scooter riders would suffer from significantly fewer head injuries along with lower severity of head injuries if using helmets was an established habit.

### Limitations

In the available data, the distance traveled on Segways and e-scooters was not documented, nor was the average speed of both groups. Hence, it was not possible to calculate the injury per mile or per speed traveled.

Furthermore, we want to mention that due to the retrospective analysis of the data, not all information was given in all detail for every single patient. Hence some clinical information may be incomplete.

## 5. Conclusions

In our study, we demonstrated the serious trauma consequences of the use of both vehicles, Segways, and e-scooters. The biggest difference that we want to point out is the fact that head and facial injuries are significantly more often observed in e-scooter injured patients. Especially regarding driving under the influence of alcohol, our data could show that the influence of alcohol during the use of e-scooters results in a significantly higher risk for a head injury. Therefore, we recommend the mandatory use of helmets for e-scooter riders. This recommendation could be strengthened by our comparison of injury patterns with mostly helmet-wearing Segway riders. This analysis demonstrates the high benefit of helmet usage in preventing serious head injuries. In addition, we are recommending a 0.0 per mile or “zero-alcohol” policy for e-scooter use as well.

## Figures and Tables

**Figure 1 medicina-58-01033-f001:**
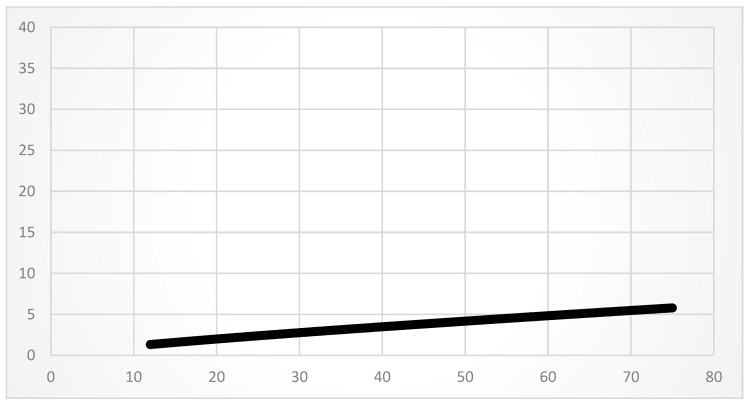
Correlation of ISS (*y*-axis) and age (*x*-axis) in non-alcoholized patients: the patients’ age is associated with more severe injury patterns.

**Table 1 medicina-58-01033-t001:** Epidemiology.

	Group I (Segway)	Group II (E-Scooter)	*p*-Value
**Total cases**	56	115	
**Age, mean ± SD (years)**	48.04 ± 14.1	33.9 ± 13.4	<0.001
**Female (%)**	39	43	0.679
	**Quantity (% [CI])**	**Quantity (% [CI])**	
**Alcohol**	0 (0 [0;0])	38 (33 [24;42])	<0.001
**Day of incident**			0.889
-Monday to Thursday	24 (43 [30;56])	48 (42 [33;51])	
-Friday to Sunday	32 (57 [44;70])	67 (58 [49;67])	
**Time of day**			
-7 a.m. to 3 p.m.	14 (25 [14;36])	22 (19 [12;26])	0.377
-3 p.m. to 9 p.m.	27 (48 [35;61])	32 (28 [20;36])	0.008
-9 p.m. to 7 a.m.	2 (4 [−1;8])	50 (43 [34;53])	<0.001
**Tourist**	30 (54 [41;67])	38 (33 [24;42])	0.01
**Commuting incident**	0 (0 [0;0])	6 (5 [1;9])	0.08
**Helmet worn**	27 (48 [35;61])	2 (1.7 [−0.7;4])	<0.001

**Table 2 medicina-58-01033-t002:** Injury patterns.

	Group I (Segway) Quantity (% [CI])	Group II (E-Scooter) Quantity (% [CI])	*p*-Value
ISS mean ± SD	6.9 ± 6.8	5.6 ± 6.4	0.211
Injury: fracture (any)	29 (52 [39;65])	51 (44 [35;53])	0.361
Injury: bruise (any)	19 (34 [22;46])	44 (38 [29;47])	0.582
Injury: wound (any)	15 (27 [15;38])	55 (48 [39;57])	0.008
Injury: dislocation (any)	2 (3 [−1;8])	4 (3 [0.1;7])	0.975
**Head**			
-Traumatic brain injury	10 (18 [8;28])	36 (31 [23;40])	0.06
-Intracranial hemorrhage	3 (5 [−0,5;11])	5 (4 [0,6;8])	0.76
-Fracture of brain skull	1 (2 [−2;5])	9 (8 [3;13])	0.11
-Fracture of viscerocranium	2 (4 [−1;8])	24 (21 [13;28])	<0.001
-Soft tissue injury	6 (11 [3;19])	33 (29 [20;37])	0.009
-Tooth fracture	0 (0 [0;0])	9 (8 [3;13])	0.03
**Upper limb**			
-Injury to upper limb	28 (50 [37;63])	44 (38 [29;47])	0.145
-Fracture of the clavicle	2 (4 [−1;8])	4 (3 [0.1;7])	0.975
-Fracture of the scapula	1 (2 [−2;5])	0 (0 [0;0])	0.151
-Fracture of the humerus	1 (2 [−1;5])	2 (2 [−0.7;4])	0.983
-Fracture of the radial head	2 (4 [−1;8])	10 (9 [4;14])	0.218
-Fracture of the olecranon	1 (1.7 [−2;5])	2 [1,7 [−0.7;4])	0.936
-Fracture of the distal radius	4 (7 [0.4;14])	4 (3 [0.1;7])	0.287
-Fracture of any hand bones	2 (4 [−1;8])	2 (1.7 [−0.7;4])	0.457
**Trunk/Pelvis**			
-Fracture of rips	3 (5 [-0.5;11])	1 (0.8 [−0.8;3])	0.068
-Fracture of the spine	2 (4 [−1;8])	1 (0.9 [−0.8;3])	0.207
-Fracture of the pelvis	3 (5 [−0.5;11])	0 (0 [0;0])	0.01
**Lower limb**			
-Injury to lower limb	16 (29 [17;40])	39 (34 [25;43])	0.483
-Fracture of the tibial head	4 (7 [0.4;14])	4 (3 [0.1;7])	0.287
-Fracture of foot/ankle	1 (2 [−2;5])	2 (1 [−0.7;4])	0.083

**Table 3 medicina-58-01033-t003:** Subgroup analysis.

	Subgroup I (E-Scooter Intoxicated) Quantity (% [CI])	Subgroup II (E-Scooter Not Intoxicated) Quantity (% [CI])	*p*-Value
**Male (%)**	29 (81 [68;93])	36(47 [36;58])	<0.001
ISS mean ± SD	7.9 ± 6.8	4.5 ± 6.4	0.004
**Injury: fracture (any)**	21 (58 [43;74])	32 (42 [31;53])	0.095
**Injury: bruise (any)**	19 (34 [22;46])	44 (38 [29;47])	0.582
**Injury: wound (any)**	25 (69 [54;84])	32 (42 [31;53])	<0.001
**Head**	34 (94 [86;102])	26 (34 [23;44])	<0.001
**Upper limb**	9 (25 [11;39])	35 (45 [34;57])	<0.001
**Lower limb**	6 (17 [4;29])	33 (43 [32;54])	<0.001
**Surgery**	12 (33 [18;49])	14 (18 [10;27])	0.075
**Inpatient submission**	14 (39 [23;55])	14 (18 [10;27])	0.017
**X-ray**	10 (27 [13;42])	63 (82 [73;90])	<0.001
**CT limbs**	0 (0 [0;0])	12 (16 [7;24])	0.012
**CT cranial/cervical spine**	24 (67 [51;82])	17 (22 [13;31])	<0.001
**CT facial skull**	23 (64 [48;80])	11 (14 [6;22])	<0.001

**Table 4 medicina-58-01033-t004:** Diagnostics/therapy in emergency room and following treatment.

	Group I (Segway) Quantity (% [CI])	Group II (E-Scooter) Quantity (% [CI])	*p*-Value
**Diagnostics**			
-X-ray	40 (71 [60;83])	70 (63 [54;71])	0.255
-CT limbs	19 (34 [22;46])	12 (10 [5;16])	<0.001
-CT cranial/cervical spine	15 (27 [15;38])	41 (36 [27;44])	0.246
-CT facial skull	2 (4 [−1;8])	34 (30 [21;38])	<0.001
**Therapy**			
-Wound treatment	6 (10 [3;19])	35 (30 [22;39])	0.004
-Splint/cast	17 (30 [18;42])	33 (29 [20;37])	0.823
-Surgery	20 (35 [23;48])	29 (25 [17;33])	0.154
-Inpatient submission	23 (41 [28;54])	27 (23 [16;31])	0.018
-Hospitalization, mean ± SD (days)	8.5 ± 11.7	6.1 ± 7.1	0.074

## Data Availability

Not applicable.

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
