# Peer review of "Is the Use of Segways or E-Scooters in Urban Areas a Potential Health Risk? A Comparison of Trauma Consequences"

_medicina, 2022, doi:10.3390/medicina58081033_

Round 1

Reviewer 1 Report

Thank you for interesting analyses.

Three suggestions:

11)      Section „Discussion” is not the best place to present „new results”. It could be better if information about  correlation of age and ISS (lines 207 – 210) were presented firstly in section “Results”, following Figure 1. Some further details could be also presented in this context.

22)      “Conclusions” - in my opinion - should firstly conclude presented in the paper scientific results. It could be a basis for some further recommendation (e.g. “a mandatory use of helmets” or” 0.0 per mile limit”) but such recommendations should not replace real, scientific conclusions. Additionally, recommendation of “a mandatory use of helmets” has not been derived from the presented results. Authors wrote: "The association between head injuries and wearing a safety helmet is obvious." - and it is probably obvious, but what evidences was presented in this paper on this thesis? Everything is clear, however, regarding negative influence of alcohol, becouse this thesis was precisely presented analysing subgroups "intoxicated e-scooter riders” and "not-intoxicated e-scooter riders".

33)      A few typos can be seen in the text, especially in "Abstract" (approval2019, Overall171)

Author Response

Dear reviewer,

thank you very much for your effort in reviewing our study. See our answers attached.

Kind regars

Dr. Kai Hoffeld

Reviewer 2 Report

Thank you for the opportunity to review this manuscript. Overall, the manuscript is concise and includes all necessary descriptions an analyses. There are some revisions that would improve accessibility and interpretation of the results:

(1) p-values are not always reported consistently. For example, Pg 4, Ln 148 uses p<0.0001 but p<.001 is used elsewhere in the text and tables. This should be consistent throughout the manuscript.

(2) It is not clear in the results when parametric and non-parametric analyses are used, though the tests are mentioned in the Discussion. Overall, the sample sizes are likely large enough to use parametric tests, it would be helpful for readers to know which subsets were non-normally distributed. 

(3) Interpretation of the Pearson-r is not clearly stated. Is r=0.23 or r=0.36 considered "strong" positive correlation for similar trauma studies? It certainly appears that there is a positive correlation between age and ISS, as is justified and explained in the Discussion, it may be spurious to claim this as a "strong" positive correlation.

(4) It is unclear what is meant on Pg. 7, Ln247: does this mean that no unique intracranial hemorrhages amongst one of the two subgroups? 

(5) The authors write well, but some general edits for redundancy, clarity, and grammar will help with for English readers. 

Author Response

(The authors gave the same response as above.)

Reviewer 3 Report

This is an interesting paper on injuries associated with the use of electric personal travel vehicles.

It describes the injuries observed in an urban setting in Germany.

Introduction

The problem is not clearly stated, explaining what this paper adds in addition to the papers quoted (references 10-14). German urban settings are already described.

A useful additional paper would be:
https://www.ncbi.nlm.nih.gov/pmc/articles/PMC8677920/

The problem is relevant, looking at segways and e-scooters, however the introduction does not explain why Segways and e-scooters have been compared together, rather than electric bikes or bicycles.

The population and risk factors are clearly identified. This paper I expect is a retrospective case series. The level of evidence and study type needs to be defined. 

It is not clear what this study will add in addition to the already stated literature which describes injuries in German cities.

Other comments:

Line 30:
Today’s cityscape is hard to imagine without them  -  needs re-wording as does not add

Line 47 ‘actual’ is superfluous

Methods

The study subjects are clearly stated. I would suggest rather than defining group I and group II, which the reader needs to remember throughout the paper, calling the groups ‘Segways’ and ‘escooters’ would be more simple.

Line 71 – ‘not unambiguously’ should be reworded

Results

The results are comprehensive, but a better selection of relevant findings should be presented. For example 3.3 Diagnostics is not clear as to what the difference in CT usage adds to the paper.
Figure 1 should be better presented with a line of best fit that obscures details.

Discussion

It is not obvious to the reader that both Segways and scooters are both available to rent and this should be highlighted in the introduction. However what proportion of scooters were privately owned as opposed to rented? Rental models are a single type of vehicle, whereas privately owned are varied in terms of power, range and speed.

This is a major flaw of this paper.

The wording throughout this paper needs to be reviewed carefully for English syntax as it can be difficult to follow.

Line 209: Excluding the source of irritation "alcoholized patient," there resulted an even stronger positive relationship between 209 age and ISS (Pearson-r 0.36).  This sentence is not clear

Another flaw is that the distance travelled on Segways and and e-scooters has not been calculated, nor the average speed of both groups so the injury per mile or per speed travelled has not been calculated. Comparing Segways and e-scooters is challenging.

https://pubmed.ncbi.nlm.nih.gov/35306193/

The discussion does not build up to the conclusion. The conclusion should be a reiteration or denouement of key points. 

Author Response

(The authors gave the same response as above.)

Reviewer 4 Report

The aim of this study was to analyze and compare the types of injuries directly related to the use of Segways and e-scooters in a major German city with over a million inhabitants. In addition, the need for medical care, the associated circumstances such as driving under the influence of alcohol, time of day and weekday were investigated.

L166: Subgroup analysis needs p for interaction and the interpretation.

L197: at first paragraph, describe the summary of the results

Results: Your results is wide. Focus on the purpose of this study.

Discussion: please don’t repeat the results.

Discussion: add the research and clinical implication.

Discussion: add the paragraphs of limitations.

Author Response

(The authors gave the same response as above.)

Round 2

Reviewer 4 Report

Accept without correction.

Author Response

thank you again for your time and effort towards our manuscript.

We changed and added your last requests and suggestions. All alterations are trackable via the “Track Changes” function and are additionally marked in yellow.
You will find the following changes in our manuscript:

We hope we have been able to implement the latest changes to your satisfaction and remain

yours sincerely

Kai Hoffeld
